# Effects of Different Net Energy Levels in Low-Protein Diversified Diets on Growth Performance, Nutrient Digestibility, Carcass Characteristics, Meat Quality, and Colonic Microbiota in Finishing Pigs

**DOI:** 10.3390/ani15182663

**Published:** 2025-09-11

**Authors:** Xintao Wang, Daiwen Chen, Junning Pu, Gang Tian, Jun He, Ping Zheng, Jie Yu, Bing Yu

**Affiliations:** 1Animal Nutrition Institute, Sichuan Agricultural University, Chengdu 611130, China; wxtzn92251@163.com (X.W.);; 2Key Laboratory of Animal Disease-Resistance Nutrition, China Ministry of Education, China Ministry of Agriculture and Rural Affairs, Key Laboratory of Sichuan Province, Chengdu 611130, China; 3Institute of Animal Nutrition, Sichuan Agricultural University and Key Laboratory for Animal Disease-Resistance Nutrition of China Ministry of Education, Chengdu 611130, China

**Keywords:** low-protein diversified rations, net energy requirement, fold model, finishing pigs

## Abstract

In China, the application of low-protein, low-soybean meal diversified feeding strategies exhibits significant importance for reducing dependence on soybean meal imports, lowering livestock and poultry feed costs, and decreasing nitrogen emissions from animal husbandry. Lowering the level of dietary protein influences energy metabolism and utilization, while the incorporation of nonconventional feed ingredients potentially alters metabolic processes and dietary energy efficiency. Consequently, feeding diversified diets requires establishing new optimal dietary net energy levels. The findings indicated that the optimal net energy for a low-protein diversified diet during the 80–100 kg phase was 9.84–10.21 MJ/kg, and 9.49–10.02 MJ/kg during the 100–130 kg phase.

## 1. Introduction

Corn and soybean meal are the primary ingredients in commercial pig rations. However, China’s heavy dependence on imported soybeans has substantially increased soybean meal cost, elevating feed expenses and raising concerns within the domestic industry. Therefore, it is crucial to explore new solutions to reduce and replace soybean meal for achieving self-sufficiency and sustainable development in the livestock industry [1]. Reducing dietary protein content is an effective strategy for reducing soybean meal dependence. Extensive research on low-protein diets demonstrated that supplementing crystalline amino acids to meet pigs’ nutritional requirements can reduce nitrogen emissions, mitigate environmental impact, and lower production costs [2]. Among gross energy (GE), digestible energy (DE), metabolizable energy (ME), and net energy (NE), only NE provides an equivalent basis for determining energy requirements and feed energy values independent of feed characteristics [3]. This highlights the importance of the NE system in meeting pigs’ nutritional needs and maximizing protein deposition [4], making it particularly suitable for formulating low-protein diets. Research on “EnziBlend Plus” supplementation in growing male pigs fed low-protein, low-energy diets showed significant impacts on blood chemistry, with specific parameters optimized at varying dietary protein and energy levels [5]. Additional research indicated that reduced protein levels affect GE digestibility, implying a corresponding effect on effective energy intake [6]. Furthermore, substantial differences in energy expenditure occur between piglets fed low-protein versus high-protein diets, which was attributed to variations in dietary-induced thermogenesis [7]. In addition, reports have indicated that feeding diets with low protein levels and appropriately reducing energy levels did not affect carcass characteristics and meat quality [8]. Fracaroli found that NE levels between 2388 and 2563 kcal/kg optimized performance in 100–130 kg pigs fed reduced crude protein (11.8%) diets, as evidenced by reduced feeding time and improved feed efficiency [9]. However, most studies on the relationship between low-protein diets and NE requirements for growing–finishing pigs are based on corn–soybean meal diets, leaving a significant research gap regarding appropriate net energy levels for low-protein diversified diets.

Additionally, various cereal grains and oilseed by-products provide high nutritional value and serve as valuable ingredients in pig feed formulations. Practical alternatives to conventional feed ingredients include broken rice, wheat, hulled barley, rapeseed meal (RSM), cottonseed meal (CSM) and peanut meal. However, RSM and CSM contain high fiber content and exhibit distinct amino acid profiles compared to soybean meal [10], differences that can significantly influence energy metabolism. For example, adding sorghum or RSM to pig diets has been shown to reduce energy digestibility [11,12]. When formulating low-protein diets with unconventional ingredients, substantial supplementation of large amounts of crystalline amino acids is often necessary to meet pigs’ amino acid requirements. Wu et al. [13] observed reduced nitrogen retention, lowered plasma amino acid concentrations, and decreased amino acid fluxes in pigs when CP was lowered and synthetic amino acids were added. Moreover, amino acids supplementation patterns can influence nitrogen metabolism, intestinal health, and energy utilization [14,15,16]. Diversified diets containing ingredients with varied chemical structures and compositions differ significantly from traditional corn–soybean meal diets. These variations may alter intestinal digestion and absorption processes [17,18], potentially requiring dietary nutrient adjustments to maintain metabolic balance. The extensive use of crystalline amino acids may also disrupt the balance between intestinal energy intake and utilization, thereby affecting energy metabolism. This emerging challenge indicates the critical need for expanded research on NE requirements.

Consequently, this study hypothesizes that two distinct dietary structures (corn–soybean meal diet versus non-corn–soybean meal diet) can significantly influence pigs’ nutritional requirements. Given this, we proposed that establishing new NE level benchmarks is essential for formulating diversified diets. This study aimed to evaluate the effect of diversified diets with varying NE levels on growth performance, apparent nutrient digestibility, carcass characteristics, meat quality, and serum biochemical indices in finishing pigs. Furthermore, the study also examined liver and muscle fat content, as well as microbial variability in colonic digesta.

## 2. Materials and Methods

This experiment was performed in accordance with the Laboratory Animal-Guideline for ethical review of animal welfare of the People’s Republic of China (GB/T 35892-2018) [19] and approved by Sichuan Agricultural University Animal Care and Use Committee.

### 2.1. Animals, Experimental Design, and Diets

A total of 108 finishing pigs (Duroc × Landrace × Yorkshire), with an average body weight of 79.8 ± 6.5 kg, were randomly assigned to six dietary treatment groups in a completely randomized design, with 6 replicates per treatment (half barrow and half female) and 3 pigs per replicate. The experiment consisted of two distinct phases. During the initial 21-day phase, pigs were fed either a corn–soybean meal diet or one of five diversified diets. The corn–soybean meal diet contained 13.5% CP and had a NE content of 10.21 MJ/kg. The diversified diets contained 11.5% CP, with NE levels ranging from 9.62 MJ/kg to 10.79 MJ/kg (Table 1). The second phase lasted for 28 days. The corn–soybean meal diet was adjusted to contain 11.3% CP and 10.02 MJ/kg NE, while the diversified diets were reformulated to contain 10.5% CP and NE levels ranging from 9.43 MJ/kg to 10.60 MJ/kg (Table 2). The corn–soybean meal diet was formulated according to the Nutrient Requirements of Swine guidelines (GB/T 39235-2020) [20].

All pigs were housed at the Teaching and Research Base of the Institute of Animal Nutrition, Sichuan Agricultural University, China. Throughout the study, animals were allowed to feed and water ad libitum, and ambient temperature was maintained at 24 ± 2 °C.

### 2.2. Growth Performance

During the experiment, finishing pigs were individually weighed on days 0, 21 and 49, while daily feed intake was accurately recorded. These data were utilized to calculate key performance indicators: average daily gain (ADG, g), average daily feed intake (ADFI, g), feed-to-gain ratio (F/G), net energy required per kilogram of weight gain (NERG, MJ/kg), and daily intake of net energy (DINE, MJ).

The F/G ratio was obtained by dividing ADFI by ADG. NERG was calculated by multiplying ADFI by the NE content per kilogram of diet then dividing by ADG. DINE was obtained by multiplying ADFI by the NE content per kilogram of diet and then dividing by 1000.

### 2.3. Sample Collection

During the final four days of each experimental period, a digestion trial (used Acid-Insoluble Ash Method) was carried out, and approximately 500 g of fresh feces were collected daily. A 10% dilute sulfuric acid solution, equivalent to 1% of the daily fecal weight, was added to each sample. Following collection, the fecal samples were pooled, dried at 65 °C for 4 h, and allowed to regain moisture. This drying and moisture-regaining cycle was repeated until a constant weight was achieved. The fecal sample was then ground, sieved (40 mesh), and stored at −20 °C for subsequent analysis. The apparent digestibility of nutrients was conducted in accordance with the procedures outlined in AOAC Official Methods of Analysis, 21st Edition (2019). The detailed method numbers are as follows: Dry matter content was determined using AOAC method 934.01. Crude protein was analyzed using AOAC method 984.13 by Kjeldahl method, with a nitrogen conversion factor of 6.25. Crude fat was determined by ether extraction following AOAC method 920.39. Gross energy was measured using a bomb calorimeter.

On days 21 and 49 of the trial, following individual weighing of all pigs, a 15 mL blood sample was collected from one pig (whose body weight was closest to the mean value of its treatment group) per pen via superior vena cava. The samples were allowed to stand at room temperature for 30 min before being centrifuged at 3500 r/min for 10 min at 4 °C. Serum samples were then collected and stored at −20 °C for further analysis.

On day 49, following blood collection, the pig was euthanized via electrical stunning to the head followed by immediate exsanguinated. Carcass-related measurements were recorded according to the Technical Regulation for Testing of Carcass Traits in Lean-type Pig (NY/T 825-2004) [21]. Parameters assessed included carcass weight, carcass length, backfat thickness, and loin muscle area. Two 50 g muscle tissue samples were excised from the longissimus dorsi (LD) muscle, while two additional 50 g tissue samples were taken from the left hepatic lobe. These samples were stored at −20 °C for fat content determination. Additionally, colonic digesta sample was collected, quickly frozen in liquid nitrogen, and stored at −80 °C until analysis.

The dressing percentage was calculated as follows: dressing percentage (%) = (weight of carcass/live weight of pig) × 100.

### 2.4. Blood Analysis

The serum, stored at −20 °C, was thawed and analyzed for the concentrations of triglyceride (TG), total cholesterol (T-CHO), alanine aminotransferase (ALT), aspartate aminotransferase (AST), high-density lipoprotein cholesterol (HDL-C), low-density lipoprotein cholesterol (LDL-C), total protein (TP), albumin (ALB), blood urea nitrogen (BUN), and glucose (GLU). All biochemical parameters were determined using commercial kits from Nanjing Jiancheng Bioengineering Institute (Nanjing, China).

### 2.5. Pork Quality

Meat quality measurement was conducted according to the Technical Code of Practice for Pork Quality Assessment (NY/T 821-2019) [22]. LD muscle samples were collected from specific anatomical locations on the left half of the carcass. LD tissue obtained from the thoracolumbar joint was used to assess meat color and marbling score. After 45 min, the muscle cross-section was measured at three random points using a precision colorimeter (NR20XE), with the average value recorded. The samples were stored at 4 °C, after 24 h, the surface layer was removed to prevent oxidation, and meat color was reassessed following the same procedure. Marbling scores were assigned using a standardized chart.

LD muscle from the penultimate 1–2 ribs was collected for Lon muscle area calculation and pH value determination. The Lon muscle area was calculated by measuring the width and height of the eye muscle with a caliper, multiplying these values, and then multiplying by 0.7. The intramuscular pH value was measured at three points on the muscle cross-section immediately after collection and again after 24 h of storage at 4 °C.

LD samples from the penultimate 3–4 ribs were prepared for assessment of drip loss and cooking loss. The muscle was cut into strips, weighed (W1), and hung in a plastic bag to prevent contact with the bag walls. After 24 h in a freezer, surface water was removed, and the weight was recorded as (W2). For cooking loss, a separate muscle sample was weighed (W3), steamed at 75–90 °C for 30 min, cooled for assessment, and reweighed (W4).

Drip loss and cooking loss were calculated using the formulas: drip loss (%) = (W1 − W2)/W1 × 100%; cooking loss (%) = (W3 − W4)/W3 × 100%.

LD tissue from the penultimate 7–9 ribs was used for shear force analysis. The whole muscle sample was cleaned of tendons, membranes, and fats and stored at 0–4 °C for 24 h. It was then heated to a core temperature of 70–75 °C, cooled, and sampled for shear force testing. The shear force measurement was performed according to the standard method with a Texture Analyser (model TA.XT.PLUS, Stable Micro Systems, Godalming, UK) equipped with a Warner–Bratzler shear blade. Prior to analysis, the muscle tissue was precisely dissected into uniform samples with dimensions of 1 cm × 1 cm × 2 cm (Width × Height × Length), ensuring the longer side was parallel to the direction of the muscle fibers to maintain consistency. Each sample was sheared at a crosshead speed of 2 mm/s. The peak force (N) required to shear the sample was obtained from the resultant force–time curve and defined as the shear force value.

### 2.6. Microbiological Diversity and Structural Analysis of Colonic Digesta

Colonic contents were obtained by slaughtering pigs, dissecting the intestines, excising the colon, and removing the contents.

16S rRNA sequencing was performed following the protocol described by Zhou et al. [23]. For genomic DNA extraction, we utilized the Zymo Research BIOMICS DNA Microprep Kit (Cat# D4301) according to the manufacturer’s instructions. The concentration of DNA samples was detected using a Tecan F200 spectrophotometer (Tecan, Männedorf, Switzerland).

The 16S rRNA primers (515F: 5′-AGAGTTTGATCATGGCTCAG-3′ and 806R: 5′-CGGTTACCTTGTTACGACTT-3′) were used for amplification by the Applied Biosystems^®^ PCR System 9700 (Thermo Fisher, Foster City, CA, USA). The cycling conditions included an initial denaturation at 98 °C for 30 s, followed by 30 cycles of denaturation at 98 °C for 5 s, annealing at 54 °C for 15 s, and extension at 72 °C for 45 s, with a final extension at 72 °C for 2 min. The PCR products were visualized on a 1% (*w*/*v*) agarose gel and purified using the Zymoclean Gel Recovery Kit (D4008). Quantification of the purified DNA was conducted with the Tecan F200. The library was constructed according to the Illumina (NEB#E7645L) library preparation protocols and then sequenced on the NovaSeq 6000 at Rhonin Biosciences Co. (Rhonin BioSciences, Chengdu, China).

Following paired-end sequence merging with FLASH, raw reads were demultiplexed by sample-specific barcodes using Sabre, with subsequent barcode excision. Quality control was then performed in QIIME2 under stringent parameters: sequences with mean Phred quality scores < 30 were filtered; reads shorter than 200 bp were eliminated; and sequences containing > 0 ambiguous bases (N) were discarded. Sequence denoising and chimera removal were executed via the Deblur algorithm within QIIME2, generating an Amplicon Sequence Variant (ASV) feature table and representative sequences. A Naïve Bayes classifier trained against the SILVA database constructed a taxonomic reference dataset, which was subsequently applied to taxonomically annotate ASV feature sequences. Multiple sequence alignment of feature sequences was performed in QIIME2, followed by phylogenetic tree construction using the integrated FastTree plugin. Finally, all samples underwent homogenization through rarefaction, resampled to the minimum sequencing depth prior to downstream analyses.

### 2.7. Short-Chain Fatty Acids of Colonic Digesta

Approximately 0.2 g of colonic digesta was homogenized with 300 μL methanol (50%), vortexed, and centrifuged. The supernatant underwent a series of dilutions and centrifugations before being stored at −20 °C. Levels of acetic acid, propionic acid, isobutyric acid, butyric acid, isovaleric acid, and valeric acid were analyzed using gas chromatography (Agilent Technologies, Wilmington, NC, USA).

### 2.8. Statistical Snalysis

The experimental unit for this experiment was the individual replicate (pen). SPSS 22.0 software (Chicago, IL, USA) was used to perform the data analysis. An independent samples T-test was used to compare the corn–soybean meal diet and the diversified diet with the same NE levels. Diversified diet groups with different NE levels were analyzed using a one-way analysis of variance (ANOVA), followed by linear regression analysis and quadratic regression analysis to identify differences among the treatments. Where significant differences were detected by ANOVA, Duncan’s multiple range test was applied for post hoc separation of treatment means. Additionally, a folding model was utilized to further analyze specific response variables. Significance was declared at *p* < 0.05.

## 3. Results

### 3.1. Growth Performance

Table 3 showed the results of growth performance of finishing pigs. During the initial phase (days 0–21), pigs on diversified diet exhibited significantly higher F/G and NERG compared to those on corn–soybean meal diet at the same NE level (*p* < 0.05). Increasing NE levels did not significantly affect most growth performance parameters (ADFI, ADG, NERG, F/G, DINE) in pigs fed diversified diets; however, linear regression analysis revealed a significant decrease in F/G as NE increased (*p* < 0.05). In the later phase (days 21–49), pigs on diversified diet exhibited an 8.6% increase in ADG compared to corn–soybean meal-fed pigs at the same NE level.

### 3.2. Apparent Total Tract Digestibility of Nutrients

As shown in Table 4, during the first phase, under constant NE level, pigs receiving a diversified diet exhibited significantly lower DM digestibility (*p* < 0.05) and significantly higher EE digestibility (*p* < 0.05) compared to those fed a corn–soybean meal diet. Additionally, CP digestibility tended to decrease (*p* = 0.09) in pigs on diversified diet. Importantly, increasing NE levels significantly enhanced the digestibility of DM, GE, EE, and CP in pigs fed diversified diets (*p* < 0.05).

Similarly, Table 4 indicated that during the second phase, at the same NE level, pigs on diversified diet demonstrated significantly higher GE, EE, and CP digestibility (*p* < 0.05) compared to those fed the corn–soybean meal-based diet. Additionally, there was tendency increased DM digestibility (*p* = 0.09). As NE levels increased, the digestibility of DM, GE, EE, and CP significantly improved in pigs fed the diversified diets. Notably, CP digestibility peaked at 10.02 MJ/kg NE before showing a subsequent decline.

### 3.3. Folding Model

After evaluating growth performance and apparent digestibility data, suitable datasets were selected for regression analysis using the fold model. The model-derived outcomes indicated that dietary NE level should be maintained at approximately 9.84 MJ/kg during the pre-finishing phase (80–100 kg), with a subsequent reduction to roughly 9.57 MJ/kg in the late finishing phase (100–130 kg) (Figure 1).

### 3.4. Serum Biochemical Indexes

At the end of the experiment, pigs receiving a corn–soybean meal diet exhibited significantly higher serum concentrations of T-CHO and AST compared to those fed a diversified diet at the same NE level (*p* < 0.05). Furthermore, in pigs consuming the diversified diets, both T-CHO and AST concentrations demonstrated a significant decrease as NE levels increased (*p* < 0.05). In contrast, ALT levels exhibited an upward trend, reaching a peak at 10.02 MJ/kg of NE before subsequently declining (*p* < 0.05). Additionally, serum GLU levels showed an increasing trend (*p* = 0.06) (Table 5).

### 3.5. Carcass Characteristics and Meat Quality

As shown in Table 6, pigs fed a corn–soybean meal diet showed a trend of improved marbling scores compared to those pigs fed a diversified diets at the same NE level (*p* = 0.09). Furthermore, in pigs consuming the diversified diets, the dressing percentage exhibited an upward trend as the NE level rose, reaching a peak at 10.02 MJ/kg (*p* < 0.05) of NE before subsequently declining. In contrast, the marbling score exhibits a trend of initial decline followed by a subsequent increase (*p* = 0.09). No other significant differences were identified in the remaining data.

### 3.6. Organ Index

Table 7 presented the results of organ index analysis. Pigs fed diversified diet exhibited significantly lower spleen index compared to those fed a corn–soybean meal diet at the same NE level (*p* < 0.05). Furthermore, with increasing NE levels, the liver index in pigs on diversified diets demonstrated an initial decrease followed by an increase change (*p* = 0.07).

### 3.7. Muscle and Liver Fat Content

At equivalent NE levels, pigs fed a diversified diet have significantly higher liver fat content than those fed a corn–soybean meal diet (*p* < 0.05). Furthermore, liver fat content increased significantly with higher NE levels, reaching a peak at 10.02 MJ/kg before declining (*p* < 0.05) in pigs fed a diversified diet. Conversely, no significant differences were observed in muscle fat content across dietary treatments (Table 8).

### 3.8. Short Chain Fatty Acid (SCFA) Concentrations in the Colonic Contents

Data on SCFA concentrations are presented in Table 9. Overall, no significant differences were observed in the overall SCFA concentrations.

### 3.9. Diversity of Colonic Microbiota

In Table 10, α-diversity of colon microbes were displayed. At the same NE level, pigs fed diversified diets had higher Chao1 and PD compared to pigs fed corn–soybean meal diets. Furthermore, PD increased with higher NE levels, reaching a peak at 10.02 MJ/kg before declining (*p* < 0.05) in pigs fed diversified diet.

The compositions of microbiota across various diets are presented in Figure 2 and Figure 3, illustrating the phylum and genus levels, respectively. At phylum level, Firmicutes, Bacteroidota, Proteobacteria, Spirochaetota, and Epsilonbacteraeota, were identified as the predominant phyla, collectively representing more than 98% of the total bacterial community across all samples. At the genus level, Prevotella 9, Prevotellaceae NK3B31 group, and several others genera accounted for over 70% of the identified groups. While no significant differences were observed at the phylum level, notable variations emerged at the genus level. Specifically, when comparing pigs fed low-protein diversified diets to those receiving normal protein corn–soybean meal diets at equivalent NE, a significant increase in the abundance of Anaerovibrio was observed in the colons of pigs comparing the low-protein diversified diet (*p* < 0.05). Furthermore, with increasing NE levels in low-protein diversified diets, Succinivibrio and Agathobacter displayed an initial decrease followed by an increase (*p* < 0.05), while Ruminococcaceae UCG-005 showed a trend towards increase (*p* = 0.06).

Species differences are illustrated in Figure 4. Compared to the normal protein corn–soybean meal diet, pigs fed low-protein diversified diet at equivalent NE levels exhibited a significant upregulation (*p* < 0.05) of several gut microbial groups in the colon, including Rikenellaceae RC9, Treponema 2, Helicobacter, and others. Conversely, Ruminococcaceae UCG-001 and Selenomonas 1 were significantly downregulated (*p* < 0.05). Furthermore, when comparing pigs fed diversified diet with low NE levels to those receiving high NE levels, several microbial groups in the colon were significantly downregulated, while others, such as Dehalobacterium and Streptococcus, were significantly upregulated (*p* < 0.05).

## 4. Discussion

The primary aim of this study was to investigate how varying levels of dietary NE in low-protein diversified diets influenced growth performance, digestibility, serum biochemistry, carcass characteristics, meat quality, and colonic microbiota of pigs. Additionally, we sought to identify the optimal NE level for pigs on a low-protein diet. The experiment revealed that F/G decreased linearly with increasing NE levels in both phases, consistent with previous research [24]. Pigs increase their feed intake of low-energy diets to meet their daily energy requirements. However, due to physical constraints such as gastrointestinal capacity, this compensation is generally incomplete, resulting in a slightly lower total energy intake compared to high-energy diets [25,26]. This suggests a strong correlation between diet energy concentration and daily feed intake [1]. Net energy required per kilogram of weight gain (NERG) did not differ significantly during the first phase; however, in the late finishing phase, a higher dietary NE concentration resulted in less NE being utilized for weight gain. Previous research has shown that increased fiber consumption elevates overall heat production [27]. In this study, the low NE group’s feed contained higher fiber content, potentially explaining the observed increased NERG. When comparing pigs fed a normal protein corn–soybean diet with those on a low-protein diversified diet, differences in NERG and F/G were observed. The higher NERG in pigs fed the diversified diet may be attributed to its higher fiber content and the higher F/G during the pre-finishing phase when fed a low-protein diet. While at equivalent NE levels, the low-protein diversified diet showed a higher F/G at the pre-finishing phase compared to the corn–soybean meal diet, this trend did not persist in the late finishing phase. This is likely due to increased tolerance for high-fiber feeds as pigs matured. The findings suggest that low-protein diversified diets may offer greater advantageous during the late finishing phase, potentially due to improved adaptation and reduced sensitivity to nutrient composition at later growth stages.

Apparent differences in digestibility are observed between two distinct weight stages of pigs: the pre-finishing phase (80–100 kg) and the late finishing phase (100–130 kg). In the pre-finishing phase, pigs fed low-protein diversified rations exhibited significantly lower digestibility of DM and CP compared to those on a normal protein corn–soybean meal diet. At the same time, GE digestibility was also marginally reduced. Interestingly, the results reversed in the late finishing phase, aligning with earlier hypotheses. He et al. reported improved digestibility and reduced F/G when a mixed meal fully replaced soybean meal, with even better results observed with partial replacement [28]. This improvement in nutrient digestibility during the late finishing phase may contribute to enhanced growth outcomes. When pigs were fed the low-protein diversified diet, nutrient digestibility increased linearly with NE levels at the pre-finishing phase. However, this trend did not continue into the late finishing phase, where nutrient digestibility did not significantly increase beyond a certain NE threshold and even showed a decrease in CP digestibility. Protein deposition increased proportionally with energy intake but plateaued at the maximum protein deposition [9]. Since the NE level in the feed reached 10.61 MJ/kg during the late finishing phase, potentially exceeding maximizing protein deposition, this could explain the stagnation in CP digestibility. To further explore the changes in NE requirements of diversified diets, we analyzed growth performance and nutrient digestibility across three parameters, allowing us to construct a folded model to determine at which NE level each parameter reached a plateau. In this model, F/G during the pre-finishing phase of pigs plateaued at 9.84 MJ/kg, while EE and CP digestibility during the late finishing phase plateaued at 9.65 MJ/kg and 9.49 MJ/kg, respectively. Therefore, we hypothesized that the lower NE levels in the diversified diet structure would adequately satisfy the requirement of pigs. However, a comparison between the values obtained from the folded model and the actual growth performance and apparent nutrient digestibility revealed that the NE level at the inflection point did not yield the optimal growth performance. This outcome may be because the model estimates a minimum inflection value. Consequently, to identify the most suitable NE level, further evaluation of other relevant indicators is needed. Such comprehensive analysis will help draw more reliable conclusions.

Changes in energy concentration can significantly influence protein deposition in pigs. In our trial, we observed a linear increase in carcass weights as NE levels increased. When comparing low-energy and high-energy diets, it is important to note that low-energy diets typically contained more fiber [29], which may lead to a heavier gastrointestinal tract weight due to the organ’s need to process indigestible materials more intensively [30]. In contrast to carcass weight, the slaughter rate demonstrated a quadratic trend which initially increased, peaking at 10.02 MJ/kg, and then declined. This pattern can be attributed to the fact that as energy approaches maximizing protein deposition, it tends to enhance fat deposition [9]. The energy levels in the diet can affect the deposition of backfat, abdominal fat, etc., and also affect intramuscular fat (IMF) [31,32]. As pigs approach this threshold in late finishing stages, they tend to store excess energy as fat, which contributes to overall body weight gain. When focusing specifically on fat deposition, subcutaneous fat deposition often takes precedence over IMF deposition [33,34]. However, some studies indicated that changes in NE levels do not significantly affect the growth performance and carcass composition of pigs [8,35]. Additionally, our study highlighted potential liver health implications. Serum indices such as AST concentration, liver index, and liver fat content indicated possible liver damage and ectopic fat deposition, particularly when pigs were fed low-energy, low-protein diversified diets [36]. Nevertheless, this study did not further explore the effects of diversified diets and NE on liver health.

Gut microbes are influenced by various factors, including energy [37], protein [38], and diet structure [39]. In this experiment, we investigated gut microbial diversity in pigs fed a low-protein diversified diet with varying the NE levels. Our findings indicate that feeding pigs a low-protein diversified diet at a constant NE level enhances colonic microbial community diversity, consistent with previous studies [1,40]. However, when low-protein diversified diets with both low and high energy levels, a decrease in diversity was observed. This contrasts with the study by Ge et al., which reported that only an increase in dietary energy reduced rumen microbial diversity [41], whereas our experiment showed a decrease at both energy levels. To adjust the NE level, we provided diets rich in fiber for the low-energy group and high in fat for the high-energy group. Previous research by Li et al. demonstrated that adding 16% wheat bran, which is high in fiber, into the diets of growing pigs, significantly increased colonic microbial diversity [42]. However, another study reported a reduction in total colonic total bacterial count when starch was replaced by oat hulls for enhancing dietary fiber content in piglets [43]. These contrasting findings let us to hypothesize that the high fiber content in the 9.44 MJ/kg group may have negatively impacted microbial diversity, whereas the lower fiber content in the 9.73 MJ/kg group could have facilitated a substantial increase, reaching its peak at 10.02 MJ/kg. Furthermore, it is well documented that high-fat diets can alter the structure of intestinal flora and reduce its diversity [44,45], which is aligns with our observation of decreased microbial diversity in the 10.31 MJ/kg and 10.61 MJ/kg groups.

Statistical analysis revealed no significance differences in bacterial abundance at the phylum level among dietary groups. However, at the same NE levels, pigs fed a low-protein diversified diet exhibited notable shifts in specific bacterial populations compared to those fed a normal protein corn–soybean meal diet. Specifically, colon samples from pigs on the low-protein diversified diet demonstrated a 66.81% reduction in Spirochaetota and a 17.23% increase in Bacteroidota. These findings align with previous reports by He et al. [1]. At the genus level, pigs on a low-protein diversified diet showed a significant upregulation of Anaerovibrio, a fiber-degrading bacterium, compared to the control group. This observation led us to hypothesize that the increased dietary fiber content in the low-protein diversified diet may have contributed to this shift [46]. Similarly, Ruminococcaceae UCG-005, a species previously implicated in obesity-related pathways [47], displayed a comparable trend in our study, correlating with indicators of liver damage observed earlier [48]. Additionally, Succinivibrio, a genus associated with the fermentation of glucose into acetic and succinic acids [49], was found to be more abundant in the stomachs of Tibetan Macaques during winter due to their high-fiber diet consumption [50]. In this study, pigs on the low-protein diversified diet, which was rich in fiber and formulated at lower energy levels, also exhibited an increase in Succinivibrio abundance. Furthermore, Agathobacter demonstrated a positive correlation with butyric acid (BA) [51], and changes in BA concentration mirrored the abundance trends of Agathobacter observed in our study. The highest abundance of Agathobacter was noted at low energy levels, possibly reflecting the influence of the high-fiber diet. In summary, multiple dominant genera displayed functional attributes associated with fiber degradation, likely as an adaptive to the elevated crude fiber content in the dietary combination. To further explore these patterns, we conducted differential species analyses.

The differential species analysis demonstrates that feeding low-protein diversified diets enhances the abundance of beneficial bacteria involved in fiber fermentation, including Rikenellaceae RC9 gut group and Treponema 2, while reducing the presence of potentially harmful bacteria such as Prevotellaceae UCG-001 and Selenomonas 1 [38,52,53,54]. Nevertheless, it is important to note that such diversified diets may also elevate the level of Helicobacter species, which could potentially increase the risk of health issues such as prolapse, diarrhea, and liver damage [55]. Conversely, feeding a low-protein diversified diet with high NE levels and increased fat content enhance the abundance of Ruminococcus 1 in the pig colon, aligning with previous research findings [56]. This diet also promotes the growth of genera that facilitate lipolysis and obesity reduction, including Bacteroides [57], Blautia [58], and Lactobacillus [59]. These beneficial bacterial species, found in the colon of pigs fed with high-energy diets, seem to be a curious phenomenon. However, studies have also revealed their presence in the intestines of mice fed with high-fat diets, which warrants further investigation [60,61]. Nevertheless, it is important to recognize that a high-fat diet can lead to an increase in the abundance of Mailhella, a harmful pathogen [62]. Additionally, diets with low energy levels and higher fiber content resulted in an elevated abundance of Dehalobacterium, which is associated with improved lipid and carbohydrate metabolism within the pig colon [63]. Such diets are also linked to blooms in fiber-degrading genera like Oscillibacter and Ruminococcaceae UCG-002, which support enhanced fiber digestion [64,65]. However, this low-NE, high-fiber diet also leads to an increase in the pathogenic bacteria Streptococcus [66].

Thus, our findings indicate that a low-protein diversified diet induces significant shifts in the colonic microbial community of pigs, favoring the proliferation of microorganisms that enhance fiber breakdown and fermentation. Comparative analysis of dietary structures revealed that a diet with lower NE level promotes colonization of beneficial microbes involved in fiber digestion, while a diet with high NE levels stimulates the growth of microorganisms that facilitate fat metabolism. This microbial modulation may contribute to a reduction in fat deposition and potentially mitigate obesity in pigs, offering a plausible explanation for the absence of significant differences in growth performance and carcass composition across different NE levels.

Based on F/G and the price per kilogram of feed, the feed cost per kilogram of body weight gain was calculated (data unpublished). Economic evaluation indicated that diversified diets did not provide a price advantage over the entire finishing phase (80–130 kg). However, a significant reduction in feed cost per kg gain was observed during the late finishing phase (80–130 kg). Under the conditions of this study, the higher overall cost may be attributed to the substantial inclusion of soybean oil for energy adjustment and the elevated market prices of the minor grains and meals used. Considering actual production conditions and ingredient market trends, the selection of feed ingredients can be strategically adjusted in response to price fluctuations to achieve better cost control. Once pigs have adapted to low-protein diversified diets, it is possible to achieve growth performance similar or superior to that provided by corn–soybean meal diets at a lower cost—a conclusion that warrants further verification.

## 5. Conclusions

In conclusion, this study demonstrated that low-protein diversified diets can effectively replace conventional corn–soybean meal diets in finishing pigs, particularly during the late finishing phase, owing to pigs’ enhanced physiological adaptation to high-fiber dietary components. Based on the folding model applied in this study and comprehensive consideration of multiple indicators, the recommended dietary NE levels are 9.84–10.21 MJ/kg for the pre-finishing phase (80–100 kg) and 9.49–10.02 MJ/kg for the late finishing phase (100–130 kg). The precise NE requirements need further investigation through studies with graded NE levels. These optimized ranges not only maintained growth performance and nutrient digestibility but also enhanced colonic microbial diversity, specifically enriching fiber-degrading and fat-metabolizing bacterial population. We recommend adopting these NE ranges in practical feeding strategies to reduce dependence on soybean meal while supporting sustainable pork production systems. It should be noted, however, that the current model was developed using a single hybrid pig genotype (Duroc × Landrace × Yorkshire), which may limit the generalizability of the findings to other breeds or genetic lines. Furthermore, although multiple meat quality parameters were evaluated, important sensory attributes such as tenderness and juiciness were not assessed—this gap restrains a comprehensive understanding of the diet’s impact on pork palatability and overall eating quality.

## Figures and Tables

**Figure 1 animals-15-02663-f001:**
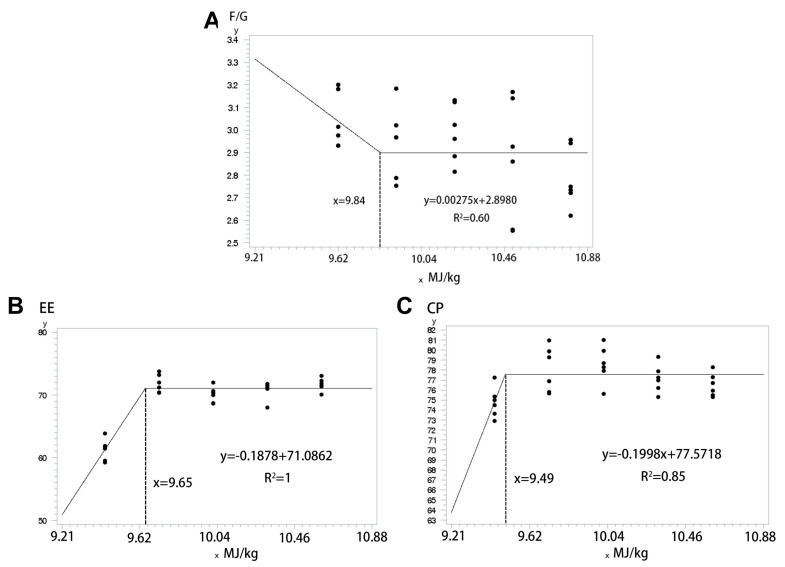
Folding model. (**A**) Folded F/G model for pigs fed low protein diversified diets with varying NE levels at the pre-finishing phase. (**B**) Folded EE digestibility model for pigs fed low protein diversified diets with varying NE levels at the late finishing phase. (**C**) Folded CP digestibility model for pigs fed low protein diversified diets with varying NE levels at the late finishing phase.

**Figure 2 animals-15-02663-f002:**
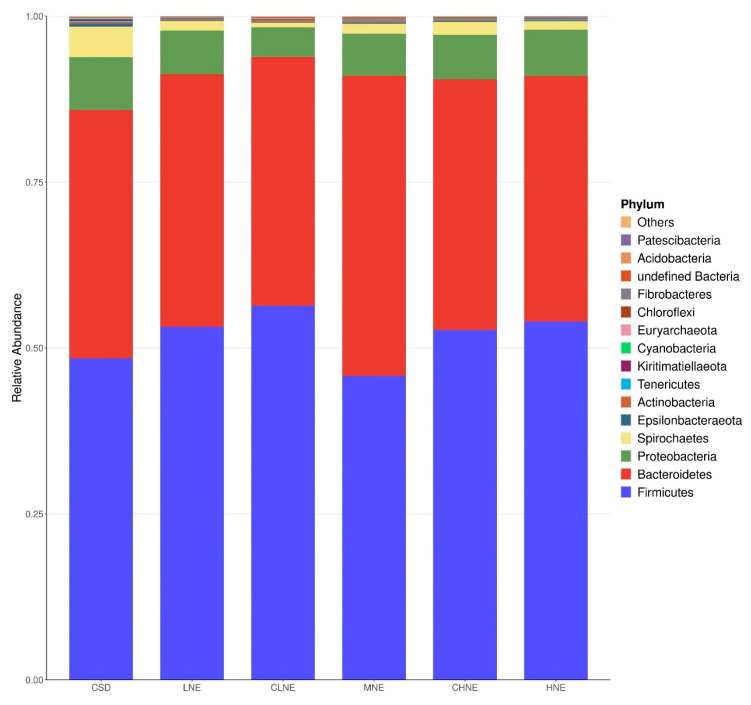
Colonic bacterial community structure on phylum level of 130 kg finishing pigs. CSD = 11.3%CP, 10.02 MJ/kg NE; LNE = 10.5%CP, 9.44 MJ/kg NE; CLNE = 10.5%CP, 9.73 MJ/kg NE; MNE = 10.5%CP, 10.02 MJ/kg NE; CHNE = 10.5%CP, 10.31 MJ/kg NE; HNE = 10.5%CP, 10.61 MJ/kg NE.

**Figure 3 animals-15-02663-f003:**
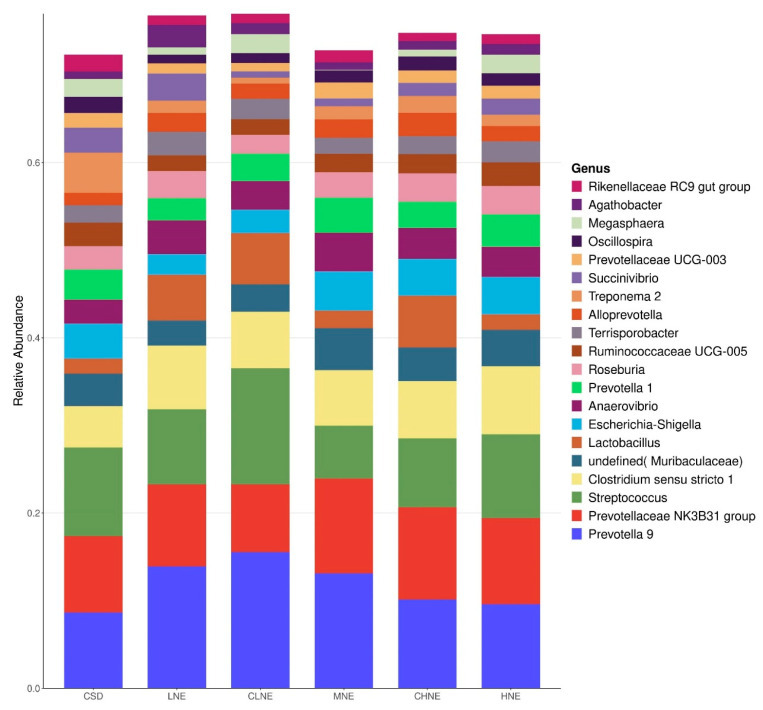
Colonic bacterial community structure on genus level of 130 kg finishing pigs. CSD = 11.3%CP, 10.02 MJ/kg NE; LNE = 10.5%CP, 9.44 MJ/kg NE; CLNE = 10.5%CP, 9.73 MJ/kg NE; MNE = 10.5%CP, 10.02 MJ/kg NE; CHNE = 10.5%CP, 10.31 MJ/kg NE; HNE = 10.5%CP, 10.61 MJ/kg NE.

**Figure 4 animals-15-02663-f004:**
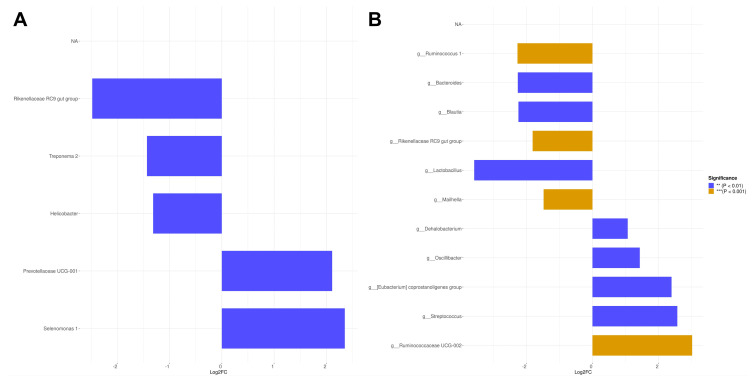
Analysis of differential species of 130 kg finishing pigs. (**A**) Analysis of species differences between CSD and MNE; CSD = corn–soybean basal diet (11.3%CP, 10.02 MJ/kg NE); MNE = diversified diets (10.5%CP, 10.02 MJ/kg NE); (**B**) Analysis of species differences between LNE and HNE; LNE = diversified diets (10.5%CP, 9.44 MJ/kg NE); HNE = diversified diets (10.5%CP, 10.61 MJ/kg NE).

**Table 1 animals-15-02663-t001:** Ingredient compositions and nutrient levels for 80–100 kg finishing pigs (%, as-fed basis).

Feed Name	13.5% CP	11.5% CP				
	10.21 MJ/kg	9.62 MJ/kg	9.92 MJ/kg	10.21 MJ/kg	10.50 MJ/kg	10.79 MJ/kg
Corn	78.60	75.21	77.08	78.96	80.83	82.70
Wheat bran	4.65	10.80	8.37	5.94	3.51	1.08
Ball milling chaff		2.46	1.85	1.23	0.62	
Rapeseed meal		3.60	4.11	4.63	5.14	5.65
Cottonseed meal		2.50	2.50	2.50	2.50	2.50
Soybean meal	13.90					
Distiller dried grains with soluble		2.00	2.00	2.00	2.00	2.00
Soybean oil			0.65	1.30	1.95	2.60
L-Lysine·HCL	0.32	0.62	0.62	0.62	0.63	0.63
DL-Methionine	0.01	0.04	0.04	0.04	0.04	0.04
L-Threonine	0.09	0.21	0.20	0.20	0.20	0.20
L-Tryptophan	0.02	0.06	0.06	0.06	0.06	0.06
L-Valine	0.02	0.15	0.15	0.15	0.15	0.15
Limestone	0.74	0.84	0.81	0.79	0.77	0.75
Chloride choline	0.15	0.15	0.15	0.15	0.15	0.15
Dicalcium phosphate	0.67	0.54	0.58	0.61	0.65	0.68
NaCl	0.30	0.30	0.30	0.30	0.30	0.30
Vitamin premix ^1^	0.03	0.03	0.03	0.03	0.03	0.03
Non-nutritive additives ^2^	0.30	0.30	0.30	0.30	0.30	0.30
Mineral premix ^3^	0.20	0.20	0.20	0.20	0.20	0.20
Total	100.00	100.00	100.00	100.00	100.00	100.00
Measured values						
Crude protein, %	14.14	12.39	12.19	11.78	12.64	11.74
Calculated values						
Crude protein, %	13.50	11.50	11.50	11.50	11.50	11.50
Metabolic energy, MJ/kg	13.10	12.26	12.59	12.93	13.22	13.56
Digestible energy, MJ/kg	13.64	12.68	13.01	13.35	13.68	14.02
Net energy, MJ/kg	10.21	9.62	9.92	10.21	10.50	10.79
Calcium, %	0.56	0.56	0.56	0.56	0.56	0.56
Available phosphorus, %	0.19	0.19	0.19	0.19	0.19	0.19
SID Lys, %	0.74	0.74	0.74	0.74	0.74	0.74
SID Met, %	0.21	0.21	0.21	0.21	0.21	0.21
SID Thr, %	0.48	0.48	0.48	0.48	0.48	0.48
SID Trp, %	0.13	0.13	0.13	0.13	0.13	0.13
SID Val, %	0.52	0.52	0.52	0.52	0.52	0.52

^1^ Vitamin premix provided the following per kg of diets: VA, 15,000 IU; VD3, 5000 IU; VE, 40.0 IU; VK3, 5.0 mg; VB1, 5.0 mg; VB2, 12.5 mg; VB6, 6.0 mg; VB12, 0.6 mg; Niacin, 50.0 mg; Pantothenic, 25.0 mg; Folic acid, 2.5 mg; Biotin, 2.5 mg. ^2^ Non-nutritive additives provided the following per kg of diets: Enzyme preparation, 0.4 g; Compound acidifier: 2.6 g. ^3^ Mineral premixes provided the following per kg of diets: Fe (FeSO_4_·H_2_O), 166.67 mg; Cu (CuSO_4_·5H_2_O), 23.90 mg; Zn (ZnSO_4_·H_2_O), 144.93 mg; Mn (MnSO_4_·H_2_O), 31.45 mg; I (KI), 7.89 mg; Se (Na_2_SeO_3_), 65.22 mg.

**Table 2 animals-15-02663-t002:** Ingredient compositions and nutrient levels for 100–130 kg finishing pigs (%, as-fed basis).

Feed Name	11.3% CP	10.5% CP				
	10.02 MJ/kg	9.44 MJ/kg	9.73 MJ/kg	10.02 MJ/kg	10.31 MJ/kg	10.61 MJ/kg
Corn	77.84	75.50	78.05	80.60	83.15	85.70
Wheat bran	13.23	8.20	6.78	5.35	3.93	2.50
Ball milling chaff		5.84	4.40	2.97	1.53	0.09
Rapeseed meal		5.00	5.00	5.00	5.00	5.00
Distiller dried grains with soluble		2.23	2.23	2.23	2.23	2.23
Soybean meal	6.00					
Soybean oil			0.30	0.60	0.90	1.20
L-Lysine·HCL	0.40	0.54	0.54	0.55	0.55	0.55
DL-Methionine	0.02	0.02	0.02	0.02	0.02	0.02
L-Threonine	0.11	0.15	0.15	0.15	0.14	0.14
L-Tryptophan	0.03	0.05	0.05	0.05	0.05	0.05
L-Valine	0.05	0.11	0.11	0.11	0.11	0.11
L-isoleucine	0.02	0.08	0.08	0.08	0.08	0.08
Limestone	0.85	0.81	0.80	0.80	0.79	0.78
Chloride choline	0.15	0.15	0.15	0.15	0.15	0.15
Dicalcium phosphate	0.47	0.50	0.52	0.54	0.56	0.58
NaCl	0.30	0.30	0.30	0.30	0.30	0.30
Vitamin premix ^1^	0.03	0.03	0.03	0.03	0.03	0.03
non-nutritive additive ^2^	0.30	0.30	0.30	0.30	0.30	0.30
Mineral premix ^3^	0.20	0.20	0.20	0.20	0.20	0.20
Total	100.00	100.00	100.00	100.00	100.00	100.00
Measured values						
Crude protein, %	11.20	10.96	10.97	11.05	10.39	10.50
Calculated values						
Crude protein, %	11.30	10.50	10.50	10.50	10.50	10.50
Metabolic energy, MJ/kg	12.77	12.02	12.36	12.69	13.03	13.37
Digestible energy, MJ/kg	13.20	12.42	12.76	13.10	13.44	13.79
Net energy, MJ/kg	10.02	9.44	9.73	10.02	10.31	10.61
Calcium, %	0.54	0.54	0.54	0.54	0.54	0.54
Available phosphorus, %	0.17	0.17	0.17	0.17	0.17	0.17
SID Lys, %	0.65	0.65	0.65	0.65	0.65	0.65
SID Met, %	0.18	0.18	0.18	0.18	0.18	0.18
SID Thr, %	0.41	0.41	0.41	0.41	0.41	0.41
SID Trp, %	0.11	0.11	0.11	0.11	0.11	0.11
SID Val, %	0.46	0.46	0.46	0.46	0.46	0.46

^1^ Vitamin premix provided the following per kg of diets: VA, 15,000 IU; VD3, 5000 IU; VE, 40.0 IU; VK3, 5.0 mg; VB1, 5.0 mg; VB2, 12.5 mg; VB6, 6.0 mg; VB12, 0.6 mg; Niacin, 50.0 mg; Pantothenic, 25.0 mg; Folic acid, 2.5 mg; Biotin, 2.5 mg. ^2^ Non-nutritive additives provided the following per kg of diets: Enzyme preparation, 0.4 g; Compound acidifier: 2.6 g. ^3^ Mineral premixes provided the following per kg of diets: Fe (FeSO_4_·H_2_O), 166.67 mg; Cu (CuSO_4_·5H_2_O), 23.90 mg; Zn (ZnSO_4_·H_2_O), 144.93 mg; Mn (MnSO_4_·H_2_O), 31.45 mg; I (KI), 7.89 mg; Se (Na_2_SeO_3_), 65.22 mg.

**Table 3 animals-15-02663-t003:** Effect of diversified diets with varying NE levels and low protein content on growth performance of finishing pigs.

**Items**	**Corn–Soybean**	**Diversified**					**SEM**	***p*-Value**			
**80–100 kg**	**13.5%CP**	**11.5%CP**						**T-Test**	**ANOVA**	**Linear**	**Quadratic**
	**10.21 MJ/kg**	**9.62 MJ/kg**	**9.92 MJ/kg**	**10.21 MJ/kg**	**10.50 MJ/kg**	**10.79 MJ/kg**					
Initial weight (kg)	79.74	79.73	79.82	79.89	79.82	79.72	1.08	0.97	1.00	1.00	1.00
Final weight (kg)	102.52	102.16	102.01	102.34	103.92	102.76	1.27	0.97	1.00	0.77	0.96
ADFI (g)	2964.68	3256.67	3118.14	3196.47	3280.74	3058.85	54.60	0.11	0.79	0.61	0.84
ADG (g)	1084.66	1067.72	1056.88	1069.31	1147.62	1096.83	15.21	0.67	0.54	0.25	0.53
F/G	2.74	3.04	2.95	2.99 *	2.87	2.79	0.03	0.03	0.12	0.01	0.04
NERG (MJ)	27.94	29.26	29.24	31.26 *	30.13	30.10	0.32	0.03	0.63	0.25	0.43
DINE (MJ)	30.27	31.34	30.92	33.41	34.45	33.02	0.57	0.11	0.47	0.14	0.32
**100–130 kg**	**11.3%CP**	**10.5%CP**					**SEM**	***p*-Value**			
	**10.02 MJ/kg**	**9.44 MJ/kg**	**9.73 MJ/kg**	**10.02 MJ/kg**	**10.31 MJ/kg**	**10.61 MJ/kg**		**T-Test**	**ANOVA**	**Linear**	**Quadratic**
Initial weight (kg)	102.52	102.16	102.01	102.34	103.92	102.76	1.27	0.97	1.00	1.00	1.00
Final weight (kg)	130.60	131.80	133.68	131.88	134.72	135.23	1.48	0.81	0.90	0.51	0.80
ADFI (g)	3419.23	3801.43	3683.56	3593.54	3738.40	3583.60	62.19	0.45	0.86	0.47	0.75
ADG (g)	1002.78	1058.13	1084.29	1089.05	1100.00	1159.92	21.14	0.08	0.56	0.10	0.25
F/G	3.41	3.71	3.40	3.41	3.33	3.17	0.06	0.57	0.83	0.04	0.13
NERG (MJ)	34.16	37.17	34.03	33.38	34.30	31.72	0.63	0.57	0.82	0.04	0.13
DINE (MJ)	34.26	35.87	35.83	36.44	38.56	38.01	0.64	0.39	0.37	0.17	0.38

Note: ADFI = Average daily feed intake; ADG = Average daily weight gain; F/G: Feed divided by weight gain; NERG: Net energy required per kilogram of weight gain; DINE: Daily intake of net energy. n = 6; T-test: the independent sample *t*-test results of corn–soybean basal diet and diversified diet with the same net energy level, * *p* < 0.05; ANOVA: One-way ANOVA analysis of different net energy levels of diversified diets.

**Table 4 animals-15-02663-t004:** Effect of diversified diets with varying NE levels and low protein content on the apparent total tract digestibility of nutrients of finishing pigs.

**Items**	**Corn–Soybean**	**Diversified**					**SEM**	***p*-Value**			
**100 kg**	**13.5%CP**	**11.5%CP**						**T-Test**	**ANOVA**	**Linear**	**Quadratic**
	**10.21 MJ/kg**	**9.62 MJ/kg**	**9.92 MJ/kg**	**10.21 MJ/kg**	**10.50 MJ/kg**	**10.79 MJ/kg**					
DM	86.04	82.51 ^d^	83.96 ^c^	84.84 *^c^	88.73 ^b^	90.72 ^a^	0.49	0.02	<0.01	<0.01	<0.01
GE	86.26	83.07 ^e^	84.39 ^d^	85.58 ^c^	89.39 ^b^	91.25 ^a^	0.49	0.15	<0.01	<0.01	<0.01
EE	68.72	74.62 ^e^	80.39 ^d^	78.81 *^c^	81.92 ^b^	85.19 ^a^	0.92	<0.01	<0.01	<0.01	<0.01
CP	82.69	80.10 ^b^	81.79 ^b^	80.89 ^b^	86.34 ^a^	87.18 ^a^	0.52	0.09	<0.01	<0.01	<0.01
**130 kg**	**11.3%CP**	**10.5%CP**					**SEM**	***p*-Value**			
	**10.02 MJ/kg**	**9.44 MJ/kg**	**9.73 MJ/kg**	**10.02 MJ/kg**	**10.31 MJ/kg**	**10.61 MJ/kg**		**T-Test**	**ANOVA**	**Linear**	**Quadratic**
DM	79.68	76.94 ^e^	79.55 ^d^	80.93 ^c^	81.88 ^b^	83.03 ^a^	0.35	0.09	<0.01	<0.01	<0.01
GE	80.19	78.07 ^e^	80.77 ^d^	82.4 *^c^	83.31 ^b^	84.47 ^a^	0.38	0.01	<0.01	<0.01	<0.01
EE	56.94	61.24 ^c^	71.83 ^a^	70.04 *^b^	70.79 ^ab^	71.69 ^ab^	1.01	<0.01	<0.01	<0.01	<0.01
CP	75.46	74.76 ^b^	78.08 ^a^	78.57 *^a^	77.14 ^a^	76.5 ^ab^	0.37	0.04	0.01	0.35	<0.01

Note: DM = dry matter; GE = gross energy; EE = ether extract; CP = crude protein. n = 6; T-test: the independent sample *t*-test results of corn–soybean basal diet and diversified diet with the same net energy level, * *p* < 0.05; ANOVA: One-way ANOVA analysis of different net energy levels of diversified diets, means with the same superscript letter within a row are not significantly different at *p* < 0.05 according to Duncan’s test.

**Table 5 animals-15-02663-t005:** Effect of diversified diets with varying NE levels and low protein content on the serum biochemical indexes of 130 kg finishing pigs.

Items	11.3%CP	10.5%CP					SEM	*p*-Value			
	10.02 MJ/kg	9.44 MJ/kg	9.73 MJ/kg	10.02 MJ/kg	10.31 MJ/kg	10.61 MJ/kg		T-Test	ANOVA	Linear	Quadratic
TG (mmol/L)	0.27	0.29	0.26	0.24	0.23	0.26	0.01	0.57	0.50	0.31	0.19
T-CHO (mmol/L)	1.93 *	1.98 ^a^	1.74 ^b^	1.71 ^b^	1.84 ^ab^	1.50 ^c^	0.04	0.05	<0.01	<0.01	0.01
AST/GOT (U/L)	6.03 *	4.55 ^b^	5.37 ^a^	3.27 ^c^	3.71 ^c^	3.59 ^c^	0.20	<0.01	<0.01	<0.01	0.01
ALT/GPT (U/L)	8.36	6.58 ^b^	7.67 ^ab^	8.89 ^a^	7.62 ^ab^	6.85 ^b^	0.24	0.59	0.06	0.81	0.02
HDL-C (mmol/L)	2.82	2.38	2.24	2.23	2.28	2.24	0.09	0.29	0.91	0.56	0.71
LDL-C (mmol/L)	0.91	0.95	0.95	0.97	0.95	0.89	0.02	0.43	0.87	0.46	0.56
TP (mg/mL)	11.81	11.90	11.90	11.99	11.89	11.91	0.02	0.06	0.47	0.92	0.67
ALB (g/L)	22.14	20.98 ^ab^	22.41 ^a^	22.47 ^a^	21.30 ^a^	19.44 ^b^	0.27	0.60	0.01	0.56	<0.01
BUN (mmol/L)	1.31	1.68	1.12	1.06	1.25	1.06	0.09	0.32	0.146	0.14	0.18
GLU (mmol/L)	3.34	3.48	3.68	3.33	3.49	4.10	0.08	0.95	0.07	0.11	0.06

Note: TG = triglyceride; T-CHO = total cholesterol; AST/GOT = glutamic oxaloacetic transaminase; ALT/GPT = glutamic-pyruvic transaminase; HDL-C = high density lipoprotein; LDL-C = low density lipoprotein; TP = total protein; ALB = albumin; BUN = blood urea nitrogen; GLU = glucose. n = 6; T-test: the independent sample *t*-test results of corn–soybean basal diet and diversified diet with the same net energy level, * *p* < 0.05; ANOVA: One-way ANOVA analysis of different net energy levels of diversified diets, means with the same superscript letter within a row are not significantly different at *p* < 0.05 according to Duncan’s test.

**Table 6 animals-15-02663-t006:** Effect of diversified diets with varying NE levels and low protein content on carcass characteristics and meat quality of 130 kg finishing pigs.

Items	11.3%CP	10.5%CP					SEM	*p*-Value			
	10.02 MJ/kg	9.44 MJ/kg	9.73 MJ/kg	10.02 MJ/kg	10.31 MJ/kg	10.61 MJ/kg		T-Test	ANOVA	Linear	Quadratic
Weight of skin (kg)	6.66	6.98	6.52	6.60	6.83	6.90	0.41	0.89	0.61	0.83	0.38
Weight of carcass (kg)	90.83	88.13	90.47	92.60	94.17	94.07	3.95	0.66	0.57	0.09	0.22
Dressing percentage (%)	69.11	67.69	68.51	70.21	69.53	69.05	0.71	0.15	0.10	0.09	0.03
Length of carcass (cm)	99.17	102.42	102.00	100.12	100.42	100.42	1.62	0.57	0.68	0.19	0.36
Average backfat thickness (cm)	1.90	2.03	2.05	2.06	1.98	1.96	0.18	0.38	0.99	0.65	0.86
Lon muscle (cm^2^)	51.22	43.59	48.89	48.04	47.69	50.28	4.10	0.46	0.52	0.16	0.34
Drip loss (%)	2.01	2.66	2.02	2.19	2.12	2.61	0.32	0.57	0.79	1.00	0.48
Cooking loss (%)	32.73	33.77	34.59	33.41	32.10	33.41	1.20	0.58	0.73	0.41	0.71
Shear force (kg·f)	7.30	7.96	6.70	7.66	6.92	7.88	1.22	0.77	0.63	0.98	0.60
Marbling score	3.67	3.67 ^a^	2.00 ^c^	2.33 ^bc^	3.17 ^ab^	2.83 ^abc^	0.70	0.09	0.01	0.69	0.09
pH_45min_	5.94	5.90	5.83	5.97	6.03	5.86	0.11	0.77	0.76	0.75	0.78
pH_24h_	5.35	5.36	5.36	5.36	5.36	5.38	0.02	0.94	0.82	0.64	0.51
L-Lightness_45min_	56.25	57.73	58.20	54.96	56.05	56.60	1.31	0.35	0.88	0.55	0.74
A-Redness_45min_	9.77	10.20	10.78	9.44	9.38	10.03	0.76	0.67	0.69	0.47	0.68
B-Yellowness_45min_	5.94	6.41	6.63	5.87	5.67	6.05	0.40	0.86	0.88	0.45	0.71
L-Lightness_24h_	71.52	70.03	71.76	69.78	72.44	72.71	1.83	0.58	0.76	0.33	0.61
A-Redness_24h_	12.78	13.20	13.43	13.53	12.02	13.14	0.35	0.20	0.18	0.33	0.63
B-Yellowness_24h_	8.17	7.68	8.12	8.12	7.42	8.53	0.32	0.94	0.20	0.38	0.62

n = 6; T-test: the independent sample *t*-test results of corn–soybean basal diet and diversified diet with the same net energy level; ANOVA: One-way ANOVA analysis of different net energy levels of diversified diets, means with the same superscript letter within a row are not significantly different at *p* < 0.05 according to Duncan’s test.

**Table 7 animals-15-02663-t007:** Effect of diversified diets with varying NE levels and low protein content on organ index of 130 kg finishing pigs.

Items	11.3%CP	10.5%CP					SEM	*p*-Value			
	10.02 MJ/kg	9.44 MJ/kg	9.73 MJ/kg	10.02 MJ/kg	10.31 MJ/kg	10.61 MJ/kg		T-Test	ANOVA	Linear	Quadratic
Heart	0.31	0.31	0.33	0.33	0.31	0.33	0.02	0.31	0.47	0.50	0.77
Liver	1.36	1.54	1.45	1.32	1.47	1.49	0.06	0.52	0.07	0.35	0.09
Spleen	0.15 *	0.14	0.15	0.12	0.14	0.13	0.01	0.01	0.10	0.34	0.62
Kidney	0.25	0.26	0.26	0.25	0.25	0.24	0.01	0.93	0.47	0.07	0.20
Leaf fat	1.15	1.25	1.33	1.42	1.33	1.45	0.16	0.13	0.85	0.35	0.63

n = 6; T-test: the independent sample *t*-test results of corn–soybean basal diet and diversified diet with the same net energy level, * *p* < 0.05; ANOVA: One-way ANOVA analysis of different net energy levels of diversified diets.

**Table 8 animals-15-02663-t008:** Effect of diversified diets with varying NE levels and low protein content on muscle and liver fat of 130 kg finishing pigs.

Items	11.3%CP	10.5%CP					SEM	*p*-Value			
	10.02 MJ/kg	9.44 MJ/kg	9.73 MJ/kg	10.02 MJ/kg	10.31 MJ/kg	10.61 MJ/kg		T-Test	ANOVA	Linear	Quadratic
Muscle fat (%)	5.39	4.38	4.49	3.30	4.41	4.51	0.15	0.11	0.48	0.89	0.54
Liver fat (%)	2.16	2.46 ^a^	2.54 ^a^	2.56 *^a^	1.90 ^b^	1.97 ^b^	0.13	0.04	<0.01	<0.01	<0.01

n = 6; T-test: the independent sample *t*-test results of corn–soybean basal diet and diversified diet with the same net energy level, * *p* < 0.05; ANOVA: One-way ANOVA analysis of different net energy levels of diversified diets, means with the same superscript letter within a row are not significantly different at *p* < 0.05 according to Duncan’s test.

**Table 9 animals-15-02663-t009:** Effect of diversified diets with varying NE levels and low protein content on concentration of short-chain fatty acids in colonic digesta in 130 kg finishing pigs.

Items	11.3%CP	10.5%CP					SEN	*p*-Value			
	10.02 MJ/kg	9.44 MJ/kg	9.73 MJ/kg	10.02 MJ/kg	10.31 MJ/kg	10.61 MJ/kg		T-Test	ANOVA	Linear	Quadratic
AA (mg/g)	9.08	9.52	9.59	8.45	8.55	10.27	0.26	0.48	0.36	0.85	0.24
PA (mg/g)	3.20	3.87	3.67	3.25	3.23	3.66	0.12	0.92	0.56	0.33	0.26
IBA (mg/g)	0.13	0.10	0.10	0.09	0.09	0.09	0.01	0.29	0.99	0.81	0.97
BA (mg/g)	1.50	2.16	1.93	1.68	1.68	2.11	0.07	0.44	0.14	0.49	0.04
IVA (mg/g)	0.26	0.20	0.21	0.20	0.20	0.22	0.01	0.20	0.96	0.57	0.80
VA (mg/g)	0.25	0.30	0.31	0.20	0.25	0.26	0.01	0.28	0.14	0.14	0.18

Note: AA = acetic acid; PA = propionic acid; IBA = isobutyric acid; BA = butyric acid; IVA = isovaleric acid; VA = valeric acid. n = 6; T-test: the independent sample *t*-test results of corn–soybean basal diet and diversified diet with the same net energy level; ANOVA: One-way ANOVA analysis of different net energy levels of diversified diets.

**Table 10 animals-15-02663-t010:** Effect of diversified diets with varying NE levels and low protein content on colonic microbial α-diversity in 130 kg finishing pigs.

Items	11.3%CP	10.5%CP					SEM	*p*-Value			
	10.02 MJ/kg	9.44 MJ/kg	9.73 MJ/kg	10.02 MJ/kg	10.31 MJ/kg	10.61 MJ/kg		T-Test	ANOVA	Linear	Quadratic
Chao1	390.11	404.30	716.98	734.14 *	570.42	495.18	43.12	0.03	0.15	0.92	0.06
Shannon	4.67	4.64	4.65	4.99	4.86	4.79	0.06	0.12	0.44	0.28	0.31
Simpson	0.97	0.97	0.95	0.98	0.98	0.98	0.00	0.24	0.12	0.28	0.53
PD	35.81	33.86	53.61	54.29 *	44.78	38.37	2.39	0.03	0.06	0.99	0.02

n = 6; T-test: the independent sample *t*-test results of a corn–soybean basal diet and a diversified diet with the same net energy levels, * *p* < 0.05; ANOVA: One-way ANOVA analysis of different net energy levels of diversified diets.

## Data Availability

The original contributions presented in this study are included in the article. Further inquiries can be directed to the corresponding author.

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
