# Peer review of "Effects of Different Net Energy Levels in Low-Protein Diversified Diets on Growth Performance, Nutrient Digestibility, Carcass Characteristics, Meat Quality, and Colonic Microbiota in Finishing Pigs"

_animals, 2025, doi:10.3390/ani15182663_

Round 1
Reviewer 1 Report
Comments and Suggestions for Authors
Dear Authors,
After careful reviewing the manuscript (animals-3821790) titled “Effects of different net energy levels in low-protein diversified diets on growth performance, nutrient digestibility, carcass characteristics, meat quality, and colonic microbiota in finishing pigs” submitted to the Animals, although the area of the study is practical and the MS is well structured, I have some comments for the authors which I believe is necessary for the MS to be scientifically solid and meeting the requirements for being published in Animals:
General comments:
- N/A
Title
- The title is well-written and informative.
Abstract and keywords
- The keywords are better to be phrases which are not already mentioned in the title.
Introduction
- The introduction in well-written, however, I believe make it more concise, particularly among L82-102, would be nice.
- The hypothesis and aim of the study are well defined.
Materials and Methods
- L116: “Duroc × Landrace × Yorkshire” instead of “Duroc× Landrace× Yorkshire”.
- L117: It is more scientific when you report mean values ± SD, reporting SD with one decimal more than the mean value. e.g. 79.8 ± 6.53.
- L151-154: These lines are redundant, as they are known simple methods of calculations in the field.
- L161: Also provide the duration of drying at this temperature.
- L163: mention the size of sieve.
- L164: provide AOAC version reference as well as method number.
- L165-168: Rewrite these lines, mentioning the methods and their corresponding codes, then provide reference for AOAC. Avoid repeating AOAC.
- L183: please mention the method of colonic digesta collection. Cite if using external methods.
- L186: “%” is redundant in the equation.
- L207-212: add more information about the used methods, while revising the final lines of the paragraph.
- L213-216: Provide more information on this method, for example dimension of samples used, the pressure in (N), and equipment used and its specifications.
- L249: provide the volume of 50% methanol used to homogenize the digesta.
Results
- L272-274: This is sort of repetition. Please merge it with Lines 268 and 269.
- Make modifications in table 3. Follow similar published papers to this end. You may better remove the second heading in the middle of the table.
- The same comment for table 4 and 5.
- L305-309: Try to use passive format. e.g. instead of “we selected suitable datasets …” you can use “Suitable datasets were selected …”. This is more scientific.
- In Figure 1A, the label of X-axis is not clear. Please consider this.
- In table footnotes explain what either of “*” and alphabets telling us.
Discussion
- In general, I recommend you to polish your discussion, removing extra sentences and words to make it more concise and scientific, without repeating results and facts in the body.
- In addition, focus more on the underlying mechanisms rather than just simply referring and comparing your results to the published literature.
- L432-433: This belongs to Results section and repeating it in Discussion is not ideal.
- It is good to clearly provide the limitations of your study at the end of this section.
Conclusion
- Same as in your discussion, the conclusion also requires polishing, and you have to provide your final recommendation in it.
References
- I see no issues in this section.
Reviewer 2 Report
Comments and Suggestions for Authors
Dear author
General comment
The manuscript addresses a highly relevant topic in swine nutrition: the optimization of net energy levels in low-protein diversified diets that partially replace conventional soybean meal. Given the increasing reliance on imported soybean meal and the need for sustainable feed alternatives, this study is timely and of practical significance. The research is well designed, with comprehensive measurements including growth performance, nutrient digestibility, serum biochemistry, carcass traits, meat quality, and gut microbiota. The findings provide novel insights and practical recommendations on optimal net energy (NE) ranges for finishing pigs. The authors examined not only performance and digestibility but also carcass characteristics, serum indices, and microbiota composition, offering a holistic view of the effects of dietary energy.
The study is suitable for publication in Animals, pending minor revisions to improve clarity, style, and interpretation.
Relevance and originality
The focus on diversified low-protein diets with different NE levels fills an important gap in the literature, as most studies rely solely on corn–soybean meal diets.
Experimental
The trial included 108 pigs, six dietary treatments, and a two-phase feeding program, ensuring sufficient replication and statistical robustness.
Clear practical implications: The identification of optimal NE ranges (10.21–10.50 MJ/kg in pre-finishing; 9.73–10.31 MJ/kg in late finishing) provides actionable recommendations for feed formulation.
Recommendation for Improvement
Economic evaluation missing: While the study highlights performance and nutrient utilization, the cost-effectiveness of diversified low-protein diets is not discussed. Since soybean replacement often aims at economic sustainability, including a basic cost analysis would strengthen applicability.
Language
The manuscript is generally well written, but there are several long sentences, repetitions, and abrupt transitions. A professional language editing would enhance readability.
Discussion
Although the results are clearly presented, the discussion could benefit from a deeper comparison with international literature, especially regarding microbiota shifts and meat quality outcomes.
Limited generalization
The trial used only one pig genotype (Duroc × Landrace × Yorkshire). The authors should acknowledge this limitation more explicitly.
Meat and carcass quality
While carcass and meat quality traits are included, no assessment of sensory or technological quality (e.g., tenderness, juiciness beyond shear force, shelf life) is provided. Mentioning this as a future research perspective would be valuable.
Recommendation
I recommend minor revision before acceptance. The manuscript is scientifically sound, methodologically rigorous, and offers useful insights for sustainable swine production. Addressing the issues above particularly improving language clarity and broadening the discussion will significantly improve its quality and impact.
Good luck,
Comments on the Quality of English LanguageThe manuscript is generally well written, but there are several long sentences, repetitions, and abrupt transitions. A professional language editing would enhance readability.
Reviewer 3 Report
Comments and Suggestions for Authors
As a final remark, the aim of the study was a. to substitute soybean meal with other protein sources and b. to check the recommended NE levels given by GB/T 39235-2020, lowering crude protein content with other protein sources but at the same time keeping SID aminoacids levels steady. Unfortunately, these guidelines (GB/T 39235-2020_ are not open accessed. Rapeseed meal and Cottonseed meal have severe antinutritional factors that can significantly affect results in pigs and in this work that aspect wsa not investigated. In my opinion the title of the manuscript, as well as the abstract should be changed to give to the reader a more precise view of the work.
Following are some more specific remarks on the paper:
Lines 17-19: You are suggesting levels of energy without specifying the type!!
Lines 15 & 21: why nonconventional feed ingredients are mentioned? See it in relation to the opening remark.
Line 54: change feed to ratios
Line 55: The cost of imported soybeans result’s in increased cost of feeding!
Line 56: Reducing protein content in diets is not a new solution for replacing soybean meal. In the next line your thoughts are better imprinted.
Line 58: The reference used is not supporting your statement.
Line 67-69: the work that you refer [5] has to do with the supplementation with “EnziBlend Plus” in growing male pigs fed low protein and low energy diets and found that it can significantly impact blood chemistry, with specific parameters optimized at different dietary protein and energy levels.
General remark: Although crude protein levels were lower the SID aninoacids levels were kept steady. This has to be mentioned in various parts of the paper.
Line 147: change with to: while
Line 158-161: It is written that 500g of fresh feces were collected daily. As I cannot see that this came out from the collection of the total amount of feces or a marker was added, I would like to have more information about this procedure.
Line 169: you should take blood samples also in the first day of the trial in order to clarify individual differences.
Line 429-431: This conclusion has a great impact on the economical outcome of the pig industry. Is there any data to understand if it is important or not on a practical basis?
Line 450: you are writing in several parts of the manuscript: the pre-finishing phase (75-100 kg) and the late finishing phase (100-125 kg). Where is the finishing phase?
Line 505: piglet should be piglets, have to check English in the whole manuscript
Comments on the Quality of English LanguageSee my previous comments. There are some spelling mistakes that have to be corrected in the whole manuscript.
Round 2
Reviewer 3 Report
Comments and Suggestions for Authors
Your revision is very good.